# Health professionals' experience on District Health Information System (DHIS2) and its utilization at local levels in Gandaki province, Nepal: A qualitative study

**Prakash Raj Bhatt**[1]*, **Rabindra Bhandari**[2], **Shiksha Adhikari**[1], **Nand Ram Gahatraj**[1]

1 Faculty of Health Sciences, School of Health and Allied Sciences, Pokhara University, Pokhara, Nepal,
2 Nepal Health Research Council, Kathmandu, Nepal

* bhattprakash070@gmail.com

**Data Availability Statement:** Data underlaying the study finding are included in the supplementary files.

## Abstract

DHIS2 is a web-based platform primarily used in developing countries, ensuring reliable data and aiding decentralized decision-making. The Ministry of Health and Population has greatly emphasized using DHIS2 for data entry and reporting. However, studies regarding health workers' experiences on DHIS2 and the utilization of data at the local level remain limited. Therefore, this study aims to investigate the usage and practical experience of DHIS2 at the local levels of Gandaki province, Nepal. An exploratory qualitative study was conducted in the Gandaki province from February to August 2023. We conducted twenty in-depth interviews among the DHIS2 users at local levels, health posts, and provincial health directorate using in-depth interview guidelines. The study participants were selected purposively. Thematic analysis was conducted to analyze the data, and NVivo was used to facilitate data analysis. Health professionals demonstrated dedication and commitment to use DHIS2 for reporting. DHIS2 has facilitated timely reporting, data storage, data analysis and visualization, feedback and communication mechanisms, and service delivery. Users' self-motivation and support from the local and provincial levels and regular review and program-specific review meetings were major facilitators for DHIS2 use. Similarly, technical issues, poor internet connectivity, power outages, and inexperienced health professionals were the significant challenges to using DHIS2. The basic and refresher training needed improvement at all levels, and learning materials were unavailable in health facilities. In addition, the data utilization at the local level in various actions was unsatisfactory despite sufficient data. Health professionals have been facilitated by DHIS2 in various actions. Capacity building of health professionals on data analysis and interpretations, continued onsite coaching, reliable internet connectivity, availability of learning materials, and improved server capacity are needed to enhance the performance of DHIS2 at the local level.

**Funding:** The author(s) received no specific funding for this work.

**Competing interests:** The authors have declared that no competing interests exist.

## Introduction

District Health Information Software (DHIS2) is an open-source, web-based platform most commonly used in Health Management Information Systems (HMIS) for collection, validation, analysis, and dissemination for all health programs at all levels of the health system [1–4]. It was developed by the University of Oslo in 1994. It is now used in over 76 Low- and Middle-Income Countries (LMICs) [3–5]. It is designed to support decentralized decision-making and health service management by allowing health workers to use their data to analyze their levels of service provision, predict service needs, and monitor and evaluate health program indicators [2,6]. So, the easy aggregation of health services-related data using DHIS2 has proven to support effective strategic planning, priority setting, and decision-making in developing countries [7]. Evidence from Bangladesh and Kenya shows that health professionals find DHIS2 more efficient than the paper-based system [1,8]. The implementation of DHIS2 has led to instant monitoring, cross-checking, setting priorities, and decision-making, which was time-consuming with the paper-based system [8]. Using DHIS2 data for MNCH information management in Sri Lanka has also improved the quality of care [9]. Shreds of evidence have also shown that users of DHIS2 face various challenges, including inexperienced health professionals, lack of technical support, infrastructure inadequacy, frequent changes to DHIS2 versions, maintaining both manual and electronic systems side-by-side, and limited use of data for local-level decision-making prevent the successful implementation of DHIS2 [2,8,10–12].

In Nepal, HMIS is the leading information platform implemented throughout the country to record health service utilization data, and it was established in 1994. Since then, it aimed to improve the quality of routinely collected data from FCHVs and community-level health workers from all health facilities across the country [13]. The Ministry of Health and Population (MoHP) introduced DHIS2 and web-based online data entry in 2014. In 2016, all district (public) health offices started HMIS online reporting through the DHIS2 platform. Similarly, in 2019, HMIS online reporting via DHIS2 platforms expanded to 753 local levels, accommodating the latest federal structure. Nearly all the local levels have started reporting through the DHIS2, while it is currently extending at the health post levels [14,15]. Gandaki province has also expanded DHIS2 at all 85 local levels and most health facilities [16]. As per the latest annual report, the on-time reporting status through DHIS2 was 74%, where online self-reporting from health facilities is about 38%, while the remaining health facilities' data are reported from local levels. At local levels, the skilled human resources for recording and reporting still need to be improved at various levels [17].

Different national health policies have mentioned the importance of health management information systems for evidence-based planning at different levels, including the local level. Similarly, the constitution of Nepal grants exclusive rights for policy formation, planning, decision-making, etc., to the local level. Consequently, there has been a significant demand for data and an improvement in the availability and use of data at the local level. Despite years of implementation as a recording and reporting tool in the HMIS of the national health system, there have been limited studies on the HMIS and DHIS2. Further, published literature indicates a limited exploration of the health professionals' experiences regarding the use of DHIS2, its data utilization, and barriers and enablers to the use of DHIS2 at the local level. To address these gaps, this study explored health professionals' experience with DHIS2 and its utilization at the local levels of Gandaki province.

## Methods

### Study setting

The study sites were the local levels of Gandaki province. It is one of the seven provinces of Nepal, located at 21,733 square kilometers. The province covers mountain, hill, and Terai

areas and has 11 districts. Out of 11 districts, three (Mustang, Manang, and Gorkha) lie in the northern part bordering China, and one (Nawalparasi East) lies in the southern part bordering India. Myagdi and Baglung lie in the western, Parbat and Syangja in the southern part, Tanahun and Lamjung in the Eastern part, and Kaski, the province's capital, is in the center [16]. This province has one metropolitan city, 26 municipalities, and 58 rural municipalities. Similarly, there are 19 government hospitals, 23 PHCCs, 490 health posts, 153 community health units, 99 urban health centers, 168 basic health service units, and 69 private hospitals. In addition, 1393 Community Outreach Clinics, 1824 immunization clinics, and 5939 Female Community Health Volunteers serve the health system [16].

## Study design

An exploratory qualitative study was employed to explore the experience of health professionals on DHIS2 and its utilization at local levels. Purposive sampling was used to select health professionals for the study. The consolidated criteria for reporting qualitative research checklist (COREQ) guided the reporting of this research [18].

## Study instrument and data collection

The Principal Investigator (PI) developed the in-depth interview guidelines with input from other authors based on the literature review and consultation with the provincial-level authorities. The guidelines were pretested among two local health professionals and finalized based on their feedback. The study population was health professionals working at the provincial and local levels (metropolitan, urban municipality, rural municipality) and health posts who use DHIS2. The health professionals from the provincial level were statistics officers, local levels were Public Health Officers (PHO), Public Health Inspectors (PHI), Health Assistants (HA), Senior Auxiliary Nurse Midwifes (Sr. ANM), Senior Auxiliary Health Workers (Sr. AHW) and health facility level were PHI, HA, Sr. ANM, Sr. AHW, Auxiliary Nurse Midwife (ANM) and Auxiliary Health Workers (AHW). Ten local levels and nine health posts were selected from the five districts of the province covering Mountain, Hill, Terai, and a health directorate office. Twenty in-depth interviews (IDIs) were carried out until data saturation was achieved [19], and each interview took approximately 20 to 40 minutes. Data was collected between June 28 2023, and July 31, 2023. All the interviews were conducted by the PRB (Master of Public Health Student trained in qualitative methodology) by visiting the selected facilities. The appropriate coordination was done through phone calls before visiting facilities to schedule the interview time. Before the start of the interview, the study's objective was explained to the respondents. IDI was conducted in Nepali, and all the interviews were audio recorded using the recorder. In addition, field notes were taken to document their impressions and support in the data analysis. We ensured confidentiality by maintaining the anonymity of the participants, with the interview data being accessible only to the research team.

## Data management and analysis

Upon the completion of each interview, the recorded audios were transcribed into Nepali and then translated into English by the authors PRB, SA, and RB. NRG cross-checked the transcripts. All authors reviewed the transcripts for any discrepancies and errors. These records, notes, transcripts, and translations were stored on a laptop and cloud while maintaining data security. Transcripts and findings were not returned to participants following analysis.

We conducted a thematic analysis based on the six steps outlined by Braun and Clarke. An inductive approach was used, which refers to the identified themes being strongly linked to the data and then moving to broader generalizations [20]. Before the coding commenced, authors

read and re-read the transcripts to familiarize them with the data and generate a basic idea about the different perspectives and experiences related to DHIS2 and its use. The generated ideas were then noted in the bullet points to understand the nature of the content in the transcripts. After generating the list of ideas, initial codes were identified in each line of the transcripts. Seventy-nine descriptive codes were identified. The mind map was prepared in an NVivo for further coding and analysis. The codes were then analyzed, grouped into similar categories, and assembled under sub-themes and main themes. The coding, candidate themes, and sub-themes development were undertaken by the primary researcher (PRB) with regular meetings and discussions with authors (SA) and (RB) and reviewed by the last author (NRG). The prepared themes, subthemes, and coding were further revised and recategorized in collaboration with all the authors. We used the software NVivo 12 to assist with data analysis.

### Ethical consideration

Ethical approval was obtained from the Institutional Review Committee (IRC) of Pokhara University (Reference Number: 140-079/080). An approval letter was also obtained from the provincial health directorate office of Gandaki province. Written informed consent was obtained from all the participants. All the interviews were carried out with voluntary participation, beginning with a clear explanation of the study's purpose. The participants were assured of their privacy and confidentiality and the information provided throughout the study process.

## Results

This qualitative study, 20 IDIs were conducted between June and July 2023, with a DHIS2 focal persons of provincial health directorate, local levels, and health posts. The demographic information for the IDI participants is illustrated in Table 1. All the study participants (n = 20) who were approached for an interview agreed to participate in the study.

### Characteristics of study participants

Table 1 shows the characteristics of study participants. Nineteen health professionals were interviewed from 5 districts of the Gandaki province and one from the provincial health directorate office. Among the participants, 12 were from hilly districts, four were from the Terai district, and three were from mountain districts. Similarly, 10 participants served as DHIS2 focal persons at local levels, nine at health posts, and one at provincial health directorate office. The study mainly consisted of male participants (n = 14). Furthermore, the majority of the participants had over three years of experience using DHIS2.

### Themes and sub-themes

The findings of the study are structured in the six themes, 1) Perceived usefulness and effectiveness of DHIS2, 2) Challenges experienced by health professionals, 3) Capacity of health professionals to use DHIS2, 4) Opportunities and potential solutions in DHIS2, 5) Experience of data utilization at the local level, and 6) Recommendations and suggestions (Table 2).

### Theme 1: Perceived usefulness and effectiveness of DHIS2

The first theme describes the perceived usefulness and effectiveness of DHIS2. It consists of four sub-themes: data entry, reporting and review, data management, maintaining data quality, and meeting conduction.

**Table 1. Characteristics of study participants.**

| Participant | District | Local level | Designation | Gender | Experience of using DHIS2 |
|---|---|---|---|---|---|
| HAPP01 | Kaski | Pokhara Metropolitan | Public Health Officer | Male | 5 years |
| HAHP02 | Kaski | | Public Health Inspector | Male | 5 Years |
| HAPP03 | Kaski | Machhapuchhre Rural Municipality | Sr. AHW[i] | Male | 5 years |
| HAHP04 | Kaski | | Sr. ANM[ii] | Female | 3 years |
| HBPP05 | Gorkha | Gorkha Municipality | Public Health Inspector | Male | 6 years |
| HBHP06 | Gorkha | | Sr. AHW | Female | 1 year |
| HBPP07 | Gorkha | Sahid Lakhan Rural Municipality | Sr. ANM | Female | 3 years |
| HBHP08 | Gorkha | | Health Assistant | Male | 2 years |
| HCPP09 | Syangja | Putalibazar Municipality | Sr. AHW | Male | 4 years |
| HCHP10 | Syangja | | Sr. AHW | Male | 3 years |
| HCPP11 | Syangja | Phedikhola Rural Municipality | Sr. ANM | Female | 3 years |
| HCHP12 | Syangja | | ANM[iii] | Female | 3 years |
| TAPP13 | Nawalparasi | Devchuli Municipality | Sr. AHW | Male | 4 years |
| TAHP14 | Nawalparasi | | Sr. AHW | Male | 2 years |
| TAPP15 | Nawalparasi | Hupsikot Rural Municipality | Sr. AHW | Male | 4 years |
| TAHP16 | Nawalparasi | | Sr. AHW | Male | 3 years |
| MAPP17 | Mustang | Gharpajung Rural Municipality | Sr. AHW | Male | 5 years |
| MAHP18 | Mustang | | AHW[iv] | Male | 1 year |
| MAPP19 | Mustang | Deloma Rural Municipality | Staff Nurse | Female | 9 months |
| PFP20 | Kaski | Provincial Health Directorate | Statistician Officer | Male | 5 years |

[I] Sr. AHW–Senior Auxiliary Health Worker.

[ii] Sr. ANM- Senior Auxiliary Nurse Midwifes.

[iii] ANM–Auxiliary Nurse midwifes.

[iv] AHW–Auxiliary Health Worker.

## Data entry, reporting, and review

At the health posts level, monthly reporting was carried out on both DHIS2 and manually. They initially prepared a monthly report on HMIS 9.3 format before entering data into DHIS2. A copy of the prepared report was also submitted to the local level. Health workers demonstrated dedication by promptly submitting reports; they even entered data from home during off hours and travelled to places with better internet and power backups.

> "... We have to enter data at the end of the month; our server gets down, and it's impossible to enter data during the day. Hence, we take all the registers at home and try to work on it at night as the Internet gets a little better then." HBPP05

After submission, the report was used to verify the entered data with hard copies to identify potential errors and discrepancies at the local level. The data review process was similar at all local levels, where during the monthly meetings, they reviewed monthly reports of all health facilities to correct errors or discrepancies in DHIS2 and paper-based reports.

## Data management

The majority of participants reported that implementing DHIS2 has greatly simplified data management and reduced errors compared to manual tallying. They emphasized that there

**Table 2. Themes and sub-themes.**

| Theme | Sub-themes |
|---|---|
| 1. Perceived usefulness and effectiveness of DHIS2 | 1.1 Data entry, reporting, and review |
| | 1.2 Data management |
| | 1.3 Maintaining data quality |
| | 1.4 Meeting conduction |
| 2. Challenges experienced by health professionals when using DHIS2 | 2.1 Technical issues |
| | 2.2 Infrastructure |
| | 2.3 Reporting locally implemented programs |
| | 2.4 Monitoring and Supervision |
| 3. Capacity of health professionals to use DHIS2 | 3.1 Self-efficacy of health professionals |
| | 3.2 Situation of training and training needs |
| | 3.3 Availability of learning materials |
| 4. Opportunities and Potential solutions in using DHIS2 | 4.1 Motivation to use DHIS2 |
| | 4.2 Self-initiative in problem-solving |
| | 4.3 Support from the health section at the local level |
| | 4.4 Support from province and district-level |
| 5. Experience in data utilization at the local level | 5.1 Data sharing practices for data utilization |
| | 5.2 Evidence-based decision making |
| | 5.3 Support and interest of authorities in data utilization |
| | 5.4 Challenges to the utilization data |
| 6. Recommendations and suggestions | |

was no need to search for physical files and that anyone could operate and utilize them from any part of the country. Users appreciated the ease of use and secure storage of data for years.

*". . . Our work has been easier due to DHIS2. . .We don't have to use our hands anymore, and finding the recording or reporting has been easier. Because of the online system, it has been easier to report. It is also easy to revise the report if there is any mistake over here and receive any feedback". HAHP02*

Additionally, participants shared that it has provided excellent assistance in data analysis and visualization, like creating presentations, bar diagrams, and charts. However, some participants found it unreliable due to system errors and preferred manual data analysis. However, many reported it was user-friendly for identifying program coverage and trends, comparing targets and achievements, and identifying gaps, which helps assess service delivery and evaluate the performance of health facilities.

## Maintaining data quality

Health professionals from both local and health post levels have witnessed the implementation of DHIS2, which has brought notable improvements in data quality. Previously, reliance on paper forms posed limitations regarding accuracy and accessibility. It had access to different validations that helped identify errors and discrepancies. Similarly, cross-checking data at the monthly meeting at the local level, as well as regular data monitoring and continuous feedback from the local, district, and province levels for timely correction, enhanced the accuracy and reliability of the data. Moreover, some local levels have social media-based messenger groups for sharing prompt information related to DHIS2.

*". . . There were mistakes when tallying by our hands. But after DHIS2, even if errors occur within DHIS2, they are monitored from the local, district, and province levels, allowing for prompt identification of the error. Therefore, the system is highly effective."* HCPP11

### Support in meeting conduction

The majority of participants reported that DHIS2 has assisted in meeting conductions like monthly, quarterly, semi-annual, and annual at the local level. Also, it is easy to generate necessary data and reports for review meetings in one click.

*"The DHIS2 tool has contributed to conducting monthly, semi-annual, and annual meetings. Instead of relying on hard-copy files, we extract the necessary data from the tool and use it for meeting discussions. It allows us quick access to the required information and facilitates analysis and review."* TAPP13

### Theme 2: Challenges experienced by health professionals

The second theme describes the experiences that the health professionals faced while using DHIS2. It consists of four subthemes: technical issues, infrastructure, challenges in reporting locally implemented programs, and monitoring and supervision.

### Technical issues

The significant difficulties highlighted by participants at both the local and health post levels were server and system-related issues. Regarding server issues, they explained that when attempting to enter monthly data during the reporting period, the servers remained busy and had to take registers home to use at night. According to the DHIS2 focal person at the provincial level, server-related problems arose due to many users. Previously, it was only accessed at district and local levels, but now, all health facilities have many users.

*"It is quite easy, but when we are in a hurry for data entry, at that time, the server goes down and makes it difficult. Also, all the health institution personnel come here for our monthly review meeting. On that day, while trying to check data of all the health institutions, DHIS2 did not function."* HCPP09

Similarly, regarding the system issues, the significant challenges faced included data lock, data loss, mismatches between data entered and reported on paper, invisible pages, and unnecessary entry warnings. Additionally, there needed to be more clarity between the demo version and the original version of DHIS2, and participants sometimes overlooked the green colour indicator during data entry, posing a risk of potential data loss. Moreover, the participants also shared that they experience issues, including mismatches between DHIS2 and paper forms, correction of mistaken data after lock, data cumulation of different indicators, and inability to enter zero values during data entry. Some participants also emphasized experiencing more problems and errors in recent days.

### Infrastructure

Internet and electricity are crucial for the efficient functioning of DHIS2. Unfortunately, most local levels and health facilities lacked good internet facilities. The study participants highlighted that the poor internet connection posed challenges in effectively using DHIS2,

even though they had access to other necessary equipment. The internet problem is due to the diverse geography of Gandaki province, where internet issues are more prevalent in rural areas than in urban areas.

*"The Internet is crucial. When the internet connection is lost while entering data, sometimes the entered data is saved, and sometimes it isn't. Similarly, the completed data are also not displayed. The Internet is available in 2–3 health facilities. Mobile data also works properly. Sometimes there is no network problem for 2–3 days, and at that time, it doesn't work."* HCPP09

Similarly, the majority of health facilities at the local level have access to electricity; however, they encounter temporary electricity issues mostly during adverse weather conditions such as rain or storms. These frequent power outages posed challenges in using DHIS2 effectively, prompting the participants to request power backups. Additionally, a few health facilities still needed more equipment, resulting in a non-functional DHIS2. Some users mentioned they only had a laptop or a desktop for operating DHIS2, but having both would be beneficial. Nevertheless, efforts were being made at the local level to provide the required equipment.

## Reporting locally implemented programs

The majority of participants at the local level shared that, since federalization, local governments have implemented various health programs using their resources. However, the DHIS2 system, designed for national priority programs, lacks the necessary tools for reporting these locally implemented programs. As a result, the data remains at the local level, making their work and contributions invisible within the health system.

*"There is no place and option to enter regarding the local government programs. It should be available as an "others" option. And we report it only at local levels; it does not report through monthly reporting."* HAPP03

Regarding these issues, the DHIS2 focal person for the province stated that the current DHIS2 system needed to include a provision to document and report all the activities carried out at the local level due to variations in program approaches across different municipalities. Although the current system managed to capture certain elements, it failed to capture all desired aspects of the program.

## Monitoring and supervision

According to the participants, monitoring and supervision practices in Gandaki province regarding DHIS2 could have been better. The majority of users from health posts shared that authorities did not visit health facilities to monitor and supervise DHIS2. Occasionally, the local level did, but only sometimes. The focal person in the province also agreed on being unable to monitor and supervise all the users; however, they regularly check data for errors and inform hundreds of users.

*"Our employees also visit district health offices, local levels, and health facilities. During these visits, they examine the service registers of the health facilities and engage in discussions with their colleagues. Additionally, if necessary, we send letters to relevant parties. The DHIS2 platform incorporates various validations to ensure data quality, and we utilize it to demonstrate*

*data quality during training sessions. Similarly, we have implemented validation rules within the DHIS2 system."* PFP20

### Theme 3: Capacity of health professionals to use DHIS2

Theme 3 describes the ability and self-efficacy of health professionals to use DHIS2 effectively at the local level. It consists of three subthemes: self-efficacy of health professionals, situation of training and training needs, and availability of learning materials.

### Self-efficacy of health professionals

The majority of the participants highlighted the noticeable deficiency of adequately trained and skilled personnel at all levels. The majority of health facilities had only one trained staff member responsible for DHIS2, which posed challenges during busy periods or when trained ones were absent. Moreover, health professionals, mainly at the health facilities level, needed to be more skilled in using pivot tables and other features of DHIS2 and faced challenges in data comparisons and visualization. Similarly, at local levels and health facilities in particular, senior staff members needed help in effectively utilizing DHIS2 due to their limited IT expertise.

> *"One significant challenge is that only one person is responsible for DHIS2 data entry in our municipality. It would have been beneficial to have additional personnel to assist in identifying errors and filling in the data."* TAPP15

### Situations of training and training needs

Most of the participants mentioned that they had received the basic training or orientation on DHIS2, and some participants shared that along with the training, they learned with their colleagues; however, a few participants, mainly from mountain districts and rural municipalities, mentioned they had been using it without training yet. Despite having received basic training, some DHIS2 users in health facilities merely entered the data without developing their data analysis and interpretation skills.

> *"No, we have not received any training this far. Even our health coordinator, who has been using DHIS2 for two years, has not received any training. None of us has been trained."* MAPP19

The study participants underlined that they had not received refresher training since their initial basic training. Moreover, they mentioned that the basic training package was short and should be comprehensive, so there will be lots of time for practical sessions and discussion to address the errors and issues. Interestingly, some local levels independently provided DHIS2 training in their areas for effective implementation.

### Availability of learning materials

The majority of participants stated that there were no hard copies of training manuals or reading materials available in health facilities and the health section at the local level. However, an e-copy of the training manual was accessible within DHIS2. Still, some participants needed to be made aware of the availability of e-copies. A focal person in the province agreed and

mentioned that there is no provision for hard copy distribution. Instead, they provide manuals or video tutorials on a pen drive as reference materials.

*"No, we were not provided with any reading material or manual. We took our laptops and learned from that. It would be better If they provided us with a Manual guide while getting any related problems or errors; we could refer to it." HCPP11*

### Theme 4: Opportunities and potential solutions in DHIS2

The third theme is opportunities and possible solutions for using DHIS2, which consists of four sub-themes, namely, motivation to use DHIS2, self-initiative in problem-solving, support from the health section at the local level, and support from the province and district level.

### Motivation to use DHIS2

The majority of health professionals were self-motivated to use DHIS2, recognizing it as something new to learn, staying updated, adapting and competing with technology, and its importance in their field. Similarly, a participant mentioned that the government's move towards digitalization motivated them to embrace it, as they wanted to stay caught up. Some participants expressed that their work motivated them to use it, as no extra benefits existed. Moreover, it allowed them to show their real-time progress and achievement and ensured accountability. Participants also cited motivations such as its data storage, analysis, visualizations, and correction capabilities, eliminating the need for paper flipping.

*"Since this is an online system and it is easy to edit. We don't have to use paper anymore; there is no need to use Tipex to correct the data. Data are easily available at any time with a single click. We can edit it from home. We can see it in different formats like pie charts and bar diagrams. It is paperless, and so is easy." HAHP02*

### Self-initiative in problem-solving

The majority of the participants shared that the first step in addressing the problem was team discussion. They tried to resolve issues based on their knowledge and skills. If they could not solve the problem themselves, they would approach the local level, primarily discussing the issues during monthly review meetings. If the problem remained unresolved, they would seek district and province-level assistance. Similarly, some participants stated that they had sought support from the Internet, including YouTube videos, and some participants entered data repeatedly until it appeared in the data set.

### Support from the health section at the local level

According to the participants, the health section at the local level played a crucial role in addressing issues faced by health facilities, where they monitored and verified reported data, corrected errors, and conducted monthly review meetings to address errors. The health section also assisted with data entry when health facility staff were unavailable and had problems with data entry. Similarly, some local levels also used a remote-controlling application (Any Desk) to facilitate problem-solving and DHIS2. However, in a metropolitan area, participants mentioned that the feedback provided from the metropolitan was not influential due to the lack of competent staff overseeing this section.

*"We do have support for issues. And if I am busy with some training or other work, they help by entering data. And if any problem is raised, they also support it."* HCHP12

Moreover, some local levels are trying to provide necessary infrastructure, such as reliable Internet, and they have allowed separate budgets specifically for Internet connectivity.

### Support from province and district-level

According to the participants, the province level played a crucial role in managing DHIS2-related issues, reviewing the reported data and feedback for timely correction to the local and health post levels. They shared feedback through various means such as emails, messages, and phone calls from the higher authorities and during review meetings like semi-annual and annual meetings. In certain districts, the district level supported health professionals utilizing DHIS2.

*"It's good. They checked the system and provided a review. We get emails from Gandaki province. They show our errors. Despite highlighting the good aspects, they highlighted our errors, which is very good for us. For example, they inquire about the accuracy of the reported data, questioning whether it was recorded correctly; if the death rate appears to be high, they investigate whether the data was genuinely entered. We get proper feedback from higher authority, which helps us improve."* HBPP05

### Theme 5: Experience in data utilization at the local level

The fifth theme is the experience of data utilization at the local level. It consists of four sub-themes: data sharing practices for data utilization, evidence-based decision-making, support and interest of authorities in data utilization, and challenges in data utilization.

### Data sharing practices for data utilization

The majority of participants reported that the data was mostly shared with decision-makers at the local level, such as elected representatives, administrative staff, health coordinators, and other required personnel. The data were shared with them through various methods at the local level. At the ward level, dissemination occurred through review meetings and presentations at the ward office. Likewise, at the local level, data were shared through review meetings, presentations, and emails.

*"We have related slides and records of the data. As necessary, data can be shown to the elected representative and administration staff and shared through email."* HCPP11

### Evidence-based decision making

The health section at the local level is the responsible body for planning and implementing health activities. According to the participants, DHIS2 has supported evidence-based decision-making for health programs. They analyzed program-specific data from DHIS2 and utilized it for effective planning. DHIS2 data was also used to prepare Annual Work Plans and Budgets (AWPB) at some local levels. Some participants also mentioned that the data have been used in resource allocation and program evaluation. However, decision-making at the local level was sometimes influenced by the interests of elected representatives rather than being evidence-based. Participants also agreed they could not fully utilize the data to the extent necessary despite the available opportunities.

*"The Gandaki province has allocated a budget for various surveillance activities in which we have utilized the DHIS2 data. Likewise, we have employed the DHIS2 data for the full immunization declaration program and similar programs. Nevertheless, we have not fully maximized the potential of utilizing data to the required extent. We have observed that certain locals have initiated commendable efforts using DHIS2 data." PFP20*

Additionally, only a few participants mentioned that the data were used for performance evaluation at the local level; however, at the province level, they used data for performance evaluation of health institutions from the district to the local level.

## Support and interest of authorities in data utilization

A mixed response was obtained in the support and interest of authorities on data utilization, where some shared that the elected representatives were unaware of the data and did not prioritize the health sector. Some mentioned that elected representatives showed interest in the data and digital programs, including paperless initiatives. According to the provincial-level focal person, they have been organizing various data literacy-related programs for elected representatives to facilitate data utilization at the local level.

*"We are conducting various data literacy programs for local elected representatives. Recently, we organized an event at the provincial level with the president and deputy president at the local level to discuss data utilization for planning. However, achieving full literacy in a day is not feasible. Developing data literacy takes time and ongoing effort. We have a dedicated health section at the local level focused on raising awareness about health data among elected representatives and stakeholders. The health workers in this section facilitate the process, and we actively engage in discussions and consultations with stakeholders during our visits." PFP20*

## Challenges to the utilization of data

The participants reported that the health sector at the local level faced various obstacles in using DHIS2 data, such as the health sector needing to be more noticed, budget prioritization for health, lack of leadership, and prevalent discrepancies in target and actual population.

*"First of all, the local level data had to be acceptable at the local level. If I take the data of any program at our local level, DHIS2 will compare the target provided from the central level based on estimation, which is not our actual data. For example, the total population is only 22394; based on this, the number of children under one year does not exceed 312. However, the target provided was 424 from the DHIS2. So, if we see the pivot table from DHIS2, it won't show our progress even if we have achieved the target of 312. Thus, data will show that health workers are not working effectively. So, we request local-level population-based data to be recognized while setting the targets. Therefore, these things should be considered during planning at the province and central level." HAPP03*

## Theme 7: Recommendations and suggestions

The seventh theme is the recommendations and suggestions provided by health professionals for the effective implementation of DHIS2.

The majority of participants from local and health post levels recommended strengthening DHIS2 at all health facilities throughout the province. They requested the implementation of electronic health records to maintain the quality of health records.

*"We need software to keep the patient profile where every patient detail is available, and that record is linked with DHIS2. If this were possible, data quality would be improved as well. We need to give attention to data quality and improve data quality". HCPP09*

The majority of participants recommended the provision of trained, competent, and adequately skilled health professionals at the health section of the local level and health facilities for the system's continuous operation. Similarly, participants suggested providing sufficient learning materials to manage the errors arising within the DHIS2.

Most participants from all levels also recommended a continuous supply of physical facilities, including electricity, internet service, and computers for timely data entry into the system. They also strongly urged the resolution of server and system issues to facilitate smooth operations within health facilities. The backing from the local level has been pivotal in the successful execution of DHIS2.

Consequently, participants suggested extending support from the local level to enhance the effective implementation of DHIS2. Additionally, study participants emphasized the importance of regular supervision and monitoring from higher levels, including backing from municipalities, provinces, and the federal level. They urged the central government to pressure local governments to enhance data quality and encourage its utilization for evidence-based planning and decision-making.

## Discussion

We aimed to explore health professionals' experiences on DHIS2 and its utilization at the local level in Gandaki province. The findings of our study identified six themes.

This study emphasizes that DHIS2 has positively influenced timely reporting, simplified reporting procedures, and contributed to health professionals' dedication and commitment toward reporting. Consistent with findings from previous studies in Nepal [13], Uganda [21], and Kenya [22], a study in Bangladesh also concluded that DHIS2 has the potential to improve the timeliness and completeness of data reporting [11]. At present, Nepalese HMIS employs both online and paper-based reporting methods. While guidelines stipulate that data entry occurs at the facility level, practical challenges, such as inadequate infrastructure and skilled health professionals, result in local health sections assisting facilities in data entry. In contrast, the data entry practice in DHIS2 varies in Pakistan [23] and Nigeria [24]. These studies reveal different data entry and reporting practices, with hard copies being submitted from health facilities to higher levels, where data entry is performed.

The successful implementation of DHIS2 at local levels encountered significant technical challenges such as server errors, data loss, data locks, invisible pages, and unnecessary entry warnings. These challenges were consistent with findings from a previous study conducted in Cameroon [25]. These system errors could be improved by enhancing the server capacity and stable and resilient Internet at all health facilities. Health professionals faced additional hurdles, including the inadequate availability of competent human resources, frequent power outages, poor internet connectivity, and insufficient infrastructure. Notably, these challenges were more pronounced in mountainous districts than in hill and Terai districts, with health professionals at health posts experiencing them to a greater extent. Similar challenges have been

documented in prior studies conducted in Nepal [13], Bangladesh [11], Cameroon [25], Lebanon [26]), Sierra Leone [27], Kenya [28], and Uganda [2].

Furthermore, monitoring and supervision from higher authorities emerged as additional obstacles to the successful implementation of DHIS2 at the local level. Health professionals are currently addressing these challenges through individual efforts, such as working from home and contacting higher authorities. As a long-term solution, prompt and coordinated efforts from all three levels of government are essential to ensure the effective implementation of DHIS2.

The training provided needed to be more comprehensive across various levels. There was significant demand for refresher training at the local level, and the need to update the existing training package was emphasized. These findings are consistent with previous studies conducted in Nepal [13,15]. Similarly, inadequate attention to training needs has been identified as a major challenge in the use of DHIS2 in other regions, including Bangladesh [11], Ghana [29], and Zanzibar [30]. A study conducted in Tanzania suggested a possible explanation for non-use, challenging the assumption that once health staff are trained, they possess the necessary skills to utilize various functions of DHIS2 [31].

To address these challenges, effective strategies observed in Kenya involved the introduction of "training of trainers" and "on-the-job training" programs for DHIS2 users [32]. Similarly, Nepal could consider incorporating DHIS2 into health-related courses, such as Proficiency Certificate Level (PCL) and public health courses, to meet training needs and enhance the utility of DHIS2. This approach aligns with the University of Colombo's initiative in Sri Lanka [12].

This study highlighted that health professionals' motivation, proficiency in data storage, analysis, visualization, and convenient data access were key factors driving the use of DHIS2. These findings resonated with a systematic review and meta-analysis, identifying technical features, effective data management, and user-friendly data access as primary strengths of DHIS2 [4]. The utilization of DHIS2 was significantly facilitated by team discussions, support from local and provincial levels, and the ongoing monitoring and feedback mechanisms implemented at the local, district, and provincial tiers. Additionally, regular review meetings and program-specific reviews at both local and provincial levels played a pivotal role in promoting the effective use of DHIS2. These findings align with similar studies conducted in Bangladesh and Sierra Leone, which identified management support, feedback loops across various levels, user enthusiasm, and the implementation of quality checks as key facilitators [11,27].

This study unveiled that the DHIS2 data were disseminated with decision makers through review meetings (monthly, semi-annual, and annual), presentations, emails, and printing hard copies. However, the data sharing primarily took place when there was a specific requirement or when someone requested. This finding is similar to previous studies conducted in Nepal [13,15].

Though we found evidence of the availability of data in DHIS2 at the local level, its utilization has yet to reach the necessary extent despite existing opportunities for utilization. The primary reasons for this suboptimal utilization include a lack of interest, low data literacy among elected representatives, and insufficient data interpretation and analysis skills. Additionally, ineffective leadership in the health sector, particularly at local levels, has been a contributing factor, often overlooked and underrated. These findings resonate with prior research conducted in Nepal [13] and Ghana [33], indicating poor utilization of data for decision-making at the local level. The health sector consistently receives lower priority than other development agendas in local-level planning and budget allocation. However, there is a notable high motivation level among health coordinators, and with additional training and support mechanisms, the state of information can be improved [13,15,33]. A systematic review in Ethiopia has

highlighted the unsatisfactory prevailing practices in effectively utilising data for decision-making, with the quality of health data still needing to be addressed [34]. Similarly, studies in Uganda and Ethiopia have identified significant barriers to data utilization, including the need for more technical capacity among health professionals for data analysis and interpretation, awareness gaps, and irregularities in supportive supervision [35,36]. The unreliability of target population data provided from the central level poses another challenge for data use at the local level, with discrepancies noted between the target population data from the central level and the actual target population at the local level.

## Limitations

The study has certain limitations. First, due to the qualitative nature of this study, the small sample size, and the fact that it was conducted in a province, the study findings may not be representative of the experiences and views of all health professionals across Nepal's health system. Second, we completely focused on the health professionals who worked at the local level. Thus, the experiences of those working below the health posts and in private health facilities were not included. Third, the study needed to incorporate stakeholders such as elected representatives and district, provincial, and national authorities, whose practical experience with DHIS2 data could provide a comprehensive understanding of its functionality. Fourth, the study was limited to exploring issues related to information security, particularly instances where health professionals need to take registers home to upload data. Nonetheless, the high levels of agreement among the study participants regarding crucial aspects may indicate the reliability of the findings. Future research should consider including public and private health facilities and engaging elected representatives and authorities at all levels of the health system to obtain a comprehensive understanding of DHIS2 and its use.

## Conclusion

In conclusion, health professionals generally viewed DHIS2 as a valuable system for efficient data management. Its implementation encountered significant challenges, including server errors, system issues, unreliable internet connections, a shortage of skilled professionals, and communication gaps between different levels. Moreover, despite ample local-level data, its utilization for evidence-based decision-making still needs improvement due to a lack of interest among elected representatives and a deficiency in the culture of data utilization. We recommend addressing server and system issues promptly from the provincial and federal levels and establishing a robust communication, monitoring, and supervision mechanism for the effective implementation of DHIS2 at the local level. Additionally, timely basic and refresher training for health professionals in DHIS2 and data management, coupled with a data literacy program for elected leaders and decision-makers at the local level, is essential for the effective implementation of DHIS2 and utilizing data for evidence-based decision-making.

## Supporting information

**S1 Checklist. COREQ checklist.**
(DOCX)

**S1 Text. IDI guide local level FP.**
(DOCX)

**S2 Text. IDI guide health post FP.**
(DOCX)

**S3 Text. IDI guide province FP.**
(DOCX)

**S4 Text. IDI transcripts local level FP.**
(DOCX)

**S5 Text. IDI transcripts health post level FP.**
(DOCX)

**S6 Text. IDI transcript province FP.**
(DOCX)

**S1 Dataset.**
(XLSX)

## Acknowledgments

The authors would like to thank all participants in this study. We would also like to thank all faculty members of Pokhara University School of Health and Allied Sciences, personnel at the health directorate, local levels, and health posts of Gandaki Province.

## Author Contributions

**Conceptualization:** Prakash Raj Bhatt, Rabindra Bhandari, Nand Ram Gahatraj.

**Data curation:** Prakash Raj Bhatt, Rabindra Bhandari, Shiksha Adhikari, Nand Ram Gahatraj.

**Formal analysis:** Prakash Raj Bhatt, Rabindra Bhandari, Shiksha Adhikari, Nand Ram Gahatraj.

**Methodology:** Prakash Raj Bhatt, Rabindra Bhandari, Nand Ram Gahatraj.

**Project administration:** Nand Ram Gahatraj.

**Software:** Prakash Raj Bhatt, Nand Ram Gahatraj.

**Supervision:** Nand Ram Gahatraj.

**Validation:** Prakash Raj Bhatt, Nand Ram Gahatraj.

**Visualization:** Prakash Raj Bhatt, Nand Ram Gahatraj.

**Writing – original draft:** Prakash Raj Bhatt, Rabindra Bhandari, Shiksha Adhikari.

**Writing – review & editing:** Prakash Raj Bhatt, Rabindra Bhandari, Shiksha Adhikari, Nand Ram Gahatraj.

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
