## [Decision Letter · Decision Letter 0]

20 Nov 2023

PGPH-D-23-01750

Health professionals experience on District Health Information System (DHIS2) and its utilization at local levels in Gandaki Province, Nepal: A qualitative study

Dear Dr. Bhatt,

Thank you for submitting your manuscript to PLOS Global Public Health. After careful consideration, we feel that it has merit but does not fully meet PLOS Global Public Health’s publication criteria as it currently stands. Therefore, we invite you to submit a revised version of the manuscript that addresses the points raised during the review process.

We look forward to receiving your revised manuscript.

Kind regards,

Carl Abelardo T. Antonio

Academic Editor

Journal Requirements:

1. In the online submission form, you indicated that "All the data are available on request". 

Additional Editor Comments (if provided):

Kindly take time to consider the comments from the reviewers. Overall, there is a need to imprpve the reporting of various parts of the manuscirpt. It might be helpful if authors can complete a qualitative study reporting guideline checklist (to be submitted as a supplementary file) to show that the minimum expected information has been included in the paper.

Reviewers' comments:

Reviewer's Responses to Questions

**Comments to the Author**

1. Does this manuscript meet PLOS Global Public Health’s publication criteria? Is the manuscript technically sound, and do the data support the conclusions? The manuscript must describe methodologically and ethically rigorous research with conclusions that are appropriately drawn based on the data presented.

Reviewer #1: Partly

Reviewer #2: Partly

2. Has the statistical analysis been performed appropriately and rigorously?

Reviewer #1: N/A

Reviewer #2: N/A

3. Have the authors made all data underlying the findings in their manuscript fully available (please refer to the Data Availability Statement at the start of the manuscript PDF file)?

Reviewer #1: Yes

Reviewer #2: No

4. Is the manuscript presented in an intelligible fashion and written in standard English?

Reviewer #1: Yes

Reviewer #2: No

5. Review Comments to the Author

Reviewer #1: With further description of their methodology, the study procedures can be reproducible. The research premise is justified and the potential to add to the literature is there. However, the authors need to show that they used a framework to guide their data collection and analysis. They also need to synthesize their findings and provide one major takeaway. They need to tie up their findings as everything seems to be all over. I'm giving them another chance at major revision to improve the structure, presentation and discussion of their findings.

Reviewer #2: 1.Introduction: There is value in concisely outlining and structuring the introduction: potentially can break paragraphs up into 1) What DHIS2 is- its history including before becoming web-based, the value to LMICs and experience in other countries, and where there is gaps in knowledge. 2) the use history and experience in Nepal (including different components that are used/customised), and where gaps in practice and knowledge exist- that led to this research need 3) what this paper and research does, and how it addresses the knowledge gap outlined in the previous two paragraphs.

2.Methods: The interviewed health professionals were purposefully sampled based on their use of the DHIS2 platform or data ie. For data entry, and decision making based on data etc and the term study population can be replaced. Also, what do the different districts mean- how are they different geographically, or socio-demographically?

3.Themes and subthemes- appears to be a tree of discussion areas mostly inspired by interview guide and questions rather than specific nuanced themes of experience or perceptions of DHIS2 utilization. For example in theme 1 , infrastructure limitation seems to be a big influence in utilization and data quality but the discussion is buried in text and not sign-posted adequately. The outside-work hour burden that is a consequence of infrastructure limitations needs to be more visible too.

4.At present, the themes are very descriptive and the research team will benefit from engaging in deeper analytical engagement with the rich data they have to convey it with more nuances in the manuscript. It is noted that a Masters student conducted interviews and was responsible for bulk of analysis. How did the other authors contribute? The manuscript and analysis will benefit from more senior academic review and input especially in writing, and analysis.

5.The paper will benefit from a process visual of the data entry and use trajectory- derived from interview findings.

6.The manuscript requires a comprehensive language check and edit- free software such as Grammarly can provide the necessary inputs.

6. PLOS authors have the option to publish the peer review history of their article (what does this mean?). If published, this will include your full peer review and any attached files.

**Do you want your identity to be public for this peer review?** For information about this choice, including consent withdrawal, please see our Privacy Policy.

Reviewer #1: **Yes: **Beryne Odeny

Reviewer #2: No

---

## [Decision Letter · Decision Letter 1]

4 Mar 2024

Health professionals experience on District Health Information System (DHIS2) and its utilization at local levels in Gandaki Province, Nepal: A qualitative study

PGPH-D-23-01750R1

Dear Mr. Bhatt,

We are pleased to inform you that your manuscript 'Health professionals experience on District Health Information System (DHIS2) and its utilization at local levels in Gandaki Province, Nepal: A qualitative study' has been provisionally accepted for publication in PLOS Global Public Health.

Best regards,

Carl Abelardo T. Antonio

Academic Editor

Thank you for taking time to address the comments raised by reviewers on your initial submission

Reviewer Comments (if any, and for reference):

Reviewer's Responses to Questions

**Comments to the Author**

1. If the authors have adequately addressed your comments raised in a previous round of review and you feel that this manuscript is now acceptable for publication, you may indicate that here to bypass the “Comments to the Author” section, enter your conflict of interest statement in the “Confidential to Editor” section, and submit your "Accept" recommendation.

Reviewer #2: All comments have been addressed

2. Does this manuscript meet PLOS Global Public Health’s publication criteria? Is the manuscript technically sound, and do the data support the conclusions? The manuscript must describe methodologically and ethically rigorous research with conclusions that are appropriately drawn based on the data presented.

Reviewer #2: Yes

3. Has the statistical analysis been performed appropriately and rigorously?

Reviewer #2: N/A

4. Have the authors made all data underlying the findings in their manuscript fully available (please refer to the Data Availability Statement at the start of the manuscript PDF file)?

Reviewer #2: No

5. Is the manuscript presented in an intelligible fashion and written in standard English?

Reviewer #2: Yes

6. Review Comments to the Author

Reviewer #2: Thank you for revising and resubmitting this manuscript. It does read much better now.

I have a minor input that are easily resolved: one more thorough read for language use and incomplete words will enhance quality- for instance, I think you mean to say terrain, and not Terai when describing the province. Please clarify.

7. PLOS authors have the option to publish the peer review history of their article (what does this mean?). If published, this will include your full peer review and any attached files.

**Do you want your identity to be public for this peer review?** For information about this choice, including consent withdrawal, please see our Privacy Policy.

Reviewer #2: No
